# Lack of an association between clinical INSTI-related body weight gain and direct interference with MC4 receptor (MC4R), a key central regulator of body weight

Carrie McMahon *, James L. Trevaskis, Christoph Carter, Kevin Holsapple,
Kirsten White, Moupali Das, Sean Collins, Hal Martin, Leigh Ann Burns-Naas

Gilead Sciences Inc., Foster City, CA, United States of America

* carrie.mcmahon@gilead.com

## Abstract

An increasing prevalence of overweight and obesity in people living with HIV has been associated with initiation of antiretroviral therapy with integrase strand transfer inhibitors (INSTIs). An off-target inhibition of the endogenous ligand binding to the human melanocortin 4 receptor (MC4R) has been suggested as a potential mechanism for clinical body weight gain following initiation of dolutegravir, an INSTI. In this study, we interrogated several INSTIs for their capacity for antagonism or agonism of MC4R in an *in vitro* cell-based assays including at concentrations far exceeding plasma concentrations reached at the recommended dosages. Our results indicate that while INSTIs do exhibit the capacity to antagonize MC4R, this occurs at concentrations well above predicted clinical exposure and is thus an implausible explanation for INSTI-associated weight gain.

## Introduction

Obesity is an increasing concern among people living with HIV (PWH). Several studies have reported an increasing prevalence of being overweight and obese in PWH, and have demonstrated that weight gain occurs in many PWH after initiating antiretroviral therapy (ART) [1–4]. Factors associated with weight gain in PWH include demographic factors (such as sex and race), HIV disease-related factors (such as disease stage and viral load), and ART-associated factors (specific antiretroviral drugs) [2, 3, 5–9]. These observations have led to several non-exclusive mechanistic hypotheses for ART-associated weight gain, including a mirroring of societal trends, a return-to-health effect of ART, improved tolerability of ART regimens, and off-target effects of antiretroviral drugs.

Among the antiretroviral drugs, the integrase strand transfer inhibitors (INSTIs) have specifically been associated with weight gain in studies of treatment-naïve PWH and in PWH switching to INSTI-based therapy [2, 7, 8, 10]. Whether this association is causative is unknown, and no mechanism to explain the association has been demonstrated. Clinical data on the effect of INSTIs on appetite has not been reported to date. An off-target effect of INSTIs has been hypothesized as a potential mechanism, based on data discussed in the European

**Data Availability Statement:** All relevant data are within the manuscript and its Supporting Information files.

**Funding:** The funder (Gilead Sciences) provided support in the form of salaries for all authors, but did not have any additional role in the study design, data collection and analysis, decision to publish, or preparation of the manuscript. The specific roles of these authors are articulated in the 'author contributions' section.

**Competing interests:** The funder (Gilead Sciences) provided support in the form of salaries for all authors, but did not have any additional role in the study design, data collection and analysis, decision to publish, or preparation of the manuscript. The specific roles of these authors are articulated in the 'author contributions' section. This does not alter our adherence to PLOS ONE policies on sharing data and materials.

Products Assessment Report for the INSTI dolutegravir (DTG), which states that DTG can inhibit the binding of endogenous ligand to the human MC4R *in vitro* [11].

The regulation of body weight is a complex, integrated process linking peripheral signals of energy stores to homeostatic responses. Key centers in the brain provide overarching control of processes that regulate food intake (via satiation and appetite, and hedonic mechanisms) and energy metabolism [12]. The prevailing overview of the central control of food intake highlights the role of the hypothalamic melanocortin system, whereby peptides derived from the precursor protein proopiomelanocortin (POMC) inhibit feeding behavior via their agonistic action on central melanocortin-3 and -4 receptors (MC3R, MC4R) [13]. Conversely, blockade of MC4R by the agouti-related protein (AgRP) increases feeding [14], and complete loss of MC3R or MC4R in mice is associated with increased food intake and concomitant obesity [15, 16]. Mutations in MC4R that render the receptor less- or non-responsive to POMC-derived peptides are commonly associated with human obesity [17]. Thus, the notion that modulation of the melanocortin system can influence food intake and body weight homeostasis is supported by rodent and clinical evidence from both genetic and pharmacological paradigms.

In this study, we have investigated the potential for approved INSTIs to interfere with endogenous ligand binding to MC4R thereby potentially providing a plausible explanation for the clinical body weight gain noted. Specifically, cellular functional assays were performed to delineate potential antagonistic or agonist effects of the following INSTIs: bictegravir (BIC), dolutegravir (DTG), cabotegravir (CAB), raltegravir (RAL), and elvitegravir (EVG). Comparisons of antagonism or agonism in the cellular assays ($IC_{50}$ values) to clinical $C_{max}$ at the recommended dosages are provided.

## Materials and methods

### Materials

Biochemical binding assays were conducted at Eurofins Cerep France and functional cellular assays were conducted at Eurofins Panlabs Discovery Services Taiwan, Ltd. Both studies were sponsored by Gilead Sciences Inc. All assay reagents and materials, including agonist reference compounds α-melanocyte stimulating hormone (α-MSH) and melanotan II and antagonist reference compounds AgRP and HS024 were obtained by the testing sites (Eurofins). Test compounds (BIC, DTG, CAB, RAL, EVG) were supplied by Gilead Sciences Inc.

### Biochemical binding assay

Binding assays were conducted to evaluate the affinity of test compounds for the human MC4R in transfected CHO cells by radioligand binding (Eurofins Cerep Catalog Item 420). Cell membrane homogenates (about 23 μg protein) were incubated for 120 min at 37˚C with 0.05 nM [125I]NDP-α-MSH in the absence or presence of the test compound in a buffer containing 25 mM Hepes/KOH (pH 7.0), 100 mM NaCl, 1.5 mM CaCl2, 1 mM MgSO4, 0.2 g/l 1.10 phenanthroline and 0.1% BSA. Nonspecific binding was determined in the presence of 1 μM NDP-α-MSH. Following incubation, the samples were filtered rapidly under vacuum through glass fiber filters (GF/B, Packard) presoaked with 0.3% PEI and rinsed several times with ice-cold 50 mM Tris-HCl using a 96-sample cell harvester (Unifilter, Packard). The filters were dried then counted for radioactivity in a scintillation counter (Topcount, Packard) using a scintillation cocktail (Microscint 0, Packard). The results are expressed as a percent inhibition of the control radioligand specific binding: 100-((measured specific binding)/(control specific binding)×100).

The standard reference compound is NDP-α-MSH, which is tested in each experiment at several concentrations to obtain a competition curve from which its $IC_{50}$ was calculated. Test

compounds were initially screened at 100 μM and run at 8 concentrations based on screening results to determine the $IC_{50}$.

## Functional cellular assays

Functional assays were conducted to evaluate the activity of test compounds for the MC4R in transfected CHO cells by measuring cAMP release using a time-resolved fluorescence resonance energy transfer (TR-FRET) method (Eurofins Panlabs Item 332270). Commercially available frozen, irradiated CHO-K1 cells with transfected human recombinant melanocortin MC4 receptor were used (PerkinElmer Part ES-191-AF). Test compound and/or vehicle was incubated with the cells (2.5 x 10E5/ml) in modified HBSS pH 7.4 buffer at 37°C for 20 minutes. The reaction was evaluated for cAMP levels by TR-FRET using a commercially available kit (PerkinElmer LANCE™ cAMP 384 kit). Test compound-induced cAMP increase by 50 percent or more (≥50%) relative to the 3 μM NDP a MSH control response indicated possible receptor agonist activity. Test compound-induced inhibition of the 10 nM NDP-a-MSH induced cAMP response by 50 percent or more (≥50%) indicated possible receptor antagonist activity. Test compounds, including concurrent known agonist and antagonist controls, were run at 6 concentrations selected based on biochemical binding assay results to determine the $IC_{50}$.

## Calculation of $IC_{50}$ values

The $IC_{50}$ values (concentration causing a half-maximal inhibition of control specific binding) and Hill coefficients (nH) were determined by non-linear regression analysis of the competition curves generated with mean replicate values using Hill equation curve fitting $Y = D + \left[ \frac{A-D}{1+(C/C_{50})^{nH}} \right]$ where Y = specific binding, A = left asymptote of the curve, D = right asymptote of the curve, C = compound concentration, C50 = $IC_{50}$, and nH = slope factor.

For biochemical binding assays, analysis was performed using software developed at Cerep (Hill software) and validated by comparison with data generated by the commercial software SigmaPlot® 4.0 for Windows® (© 1997 by SPSS Inc.). For functional cellular assays, analysis was performed using MathIQ™ (ID Business Solutions Ltd., UK).

## Results

The objective of an initial screening study was to evaluate the potential for various INSTIs to interfere with ligand binding to the MC4R in an *in vitro* biochemical assay. Specifically, binding was calculated as % inhibition of the binding of a radioactively labeled ligand (NDP-α-MSH), specific for MC4R, at a fixed concentration of each INSTI (100 μM). Any result showing an inhibition greater than 50% was considered to represent a significant effect and followed-on by determination of $IC_{50}$ values using a multi-dose concentration curve. The initial biochemical binding assay at a single 100 μM concentration showed a range from 55% to 91% binding inhibition for the five INSTIs evaluated (S1 Table). Specifically, in this study DTG showed 91% inhibition at 100 μM compared to the previously reported inhibition binding value of 64% at 10 μM [11]. Given the significant binding for all compounds assayed (>50%), $IC_{50}$ values were determined, and ranged from 0.46 μM to 78 μM.

To further evaluate the biochemical findings in a more physiologically relevant system, we performed cell-based *in vitro* MC4R antagonism and agonism assays. Validity and specificity of the assays were established with the concurrent inclusion of MC4R agonist or antagonist positive controls (Table 1). Positive controls for antagonism included AgRP or HS024 or SHU9119, with potent effects at relatively low concentrations in alignment with historical data

**Table 1. Summary of MC4R agonism and antagonism assay results and clinical $C_{max}$ margins for INSTIs.**

| Compound | Agonist | | Antagonist | |
|---|---|---|---|---|
| | $IC_{50}$ (µM) | Fold-margin on human $C_{max}$ (unbound)[a] | $IC_{50}$ (µM) | Fold-margin on human $C_{max}$ (unbound)[a] |
| BIC | >100 | >2900 | >100 | >2900 |
| DTG | >100 | >1600 | 69.4 | 1120 |
| CAB | >30 | >1200 | >30 | >1200 |
| EVG | >300 | >13000 | 2.75 | 120 |
| RAL | >300 | >160 | >300 | >160 |
| AgRP | - | - | 0.012 | NA |
| HS024 | - | - | 0.23 | NA |
| SHU9119 | - | - | 0.0069[b] | NA |
| α-MSH | 0.49 | NA | - | - |
| MT II | 0.0054 | NA | - | - |
| NDP-α-MSH | 1.67[b] | NA | - | - |

'NA'–data not available.

a Estimated margin based upon clinical $C_{max}$ (unbound) for BIC (0.034 µM), DTG (0.062 µM), CAB (0.025 µM), EVG (0.0225 µM), or RAL (1.9 µM) based upon clinical dose.

b Mean based upon on multiple independent determinations.

(per vendor confirmation). Positive controls for agonism included the endogenous agonist αMSH, MTII, and NDP-α-MSH which likewise showed selective results. Overall, concurrent positive controls for antagonism or agonism confirmed the assays performed as anticipated.

INSTIs evaluated in the functional cellular *in vitro* assay for potential antagonistic or agonistic effects are listed in Table 1. Exposure margins comparing the *in vitro* assay to the human clinical exposures were determined by dividing the $IC_{50}$ values by the clinical $C_{max}$ (unbound; protein-free) values based upon maximum approved clinical dose (S2 Table). It is noted that for BIC, DTG, and CAB, the highest concentration of INSTI evaluated was limited by compound solubility in the cellular buffer solution. However, overall, no agonistic effects were observed at any concentration of INSTIs evaluated; $IC_{50}$ values were not able to be determined up to the highest concentrations tested. As a result, exposure margins for MC4R agonism based upon $IC_{50}$ values were greater than 160- to 13,000-fold versus the clinical $C_{max}$ (unbound) values. For the antagonism assay, $IC_{50}$ values were calculated for DTG (69.4 µM) or EVG (2.75 µM) resulting in $C_{max}$ margins of 1120- or 120-fold, respectively. The remaining INSTIs evaluated, BIC, CAB, and RAL, were tested up to maximum dose concentrations of 100, 30, or 300 µM (based upon solubility for BIC or CAB), respectively, and $IC_{50}$ values were unable to be determined, resulting in $C_{max}$ margins of greater than 2900-, 1200-, or 160-fold, respectively. Fig 1 further depicts the considerable margin between $IC_{50}$ calculations and $C_{max}$ from the clinical dose for antagonistic effects.

## Discussion

Due to the potential off-target liability of MC4R for body weight gain cited for DTG, and observed body weight gain in patients receiving INSTIs, a number of INSTIs were evaluated for potential interaction with MC4R in cellular functional antagonist or agonist assays. Overall, our data demonstrate that all tested INSTIs have the ability to antagonize the MC4R receptor to some degree, but importantly, inhibition occurs only at drug concentrations substantially greater than the therapeutic plasma concentrations of each drug, ranging from 160-fold to >13,000-fold greater than unbound $C_{max}$. Furthermore, the relative $IC_{50}$ values for the INSTIs tested are inconsistent with their relative risk for clinical weight gain. For instance, DTG is

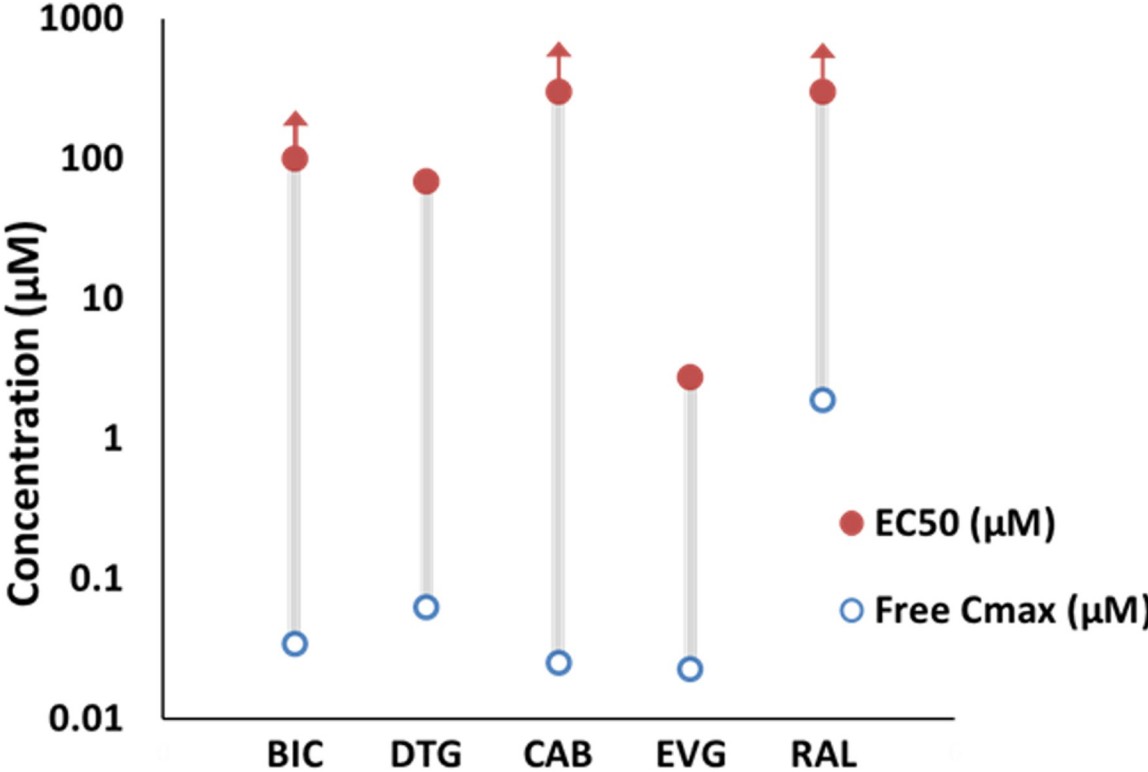

**Fig 1. Depiction of MC4R antagonism assay results and clinical $C_{max}$ margins for INSTIs.** Lines depict the substantial difference between the clinical dose $C_{max}$ value (protein-free) and the reported $EC_{50}$ value in the MC4R antagonism assay for each INSTI evaluated.

associated with greater weight gain than EVG [18], however EVG was approximately 25-fold more potent in MC4R antagonism than DTG in our experiments. These findings suggest that it is unlikely that MC4R antagonism contributes to the weight gain associated with INSTI use.

There are several limitations to our findings. Our data are derived from an *in vitro* assay, and thus do not recreate the physiological milieu in which a potential interaction between INSTIs and MC4R would occur *in vivo*. For instance, because MC4R is localized primarily to the hypothalamus, the ratio of MC4R $IC_{50}$ to plasma free drug concentrations may not be physiologically relevant. However, studies using cerebrospinal fluid INSTI concentration as a surrogate for central nervous system penetration have documented CSF drug levels equal to or less than free plasma levels, making it unlikely that more free drug is available for interaction with MC4R in the central nervous system [19–21]. It is noted that following daily oral administration with BIC in repeat dose toxicity studies, no effects on body weight gain in mice (4 weeks), rats (26 weeks), or cynomolgous monkeys (39 weeks) were reported with AUC plasma exposure levels at 16- to 30-fold the BIC plasma exposure at the clinical dose of 50 mg in Biktarvy®. While we do not know what fractional antagonism of MC4R is needed to induce weight gain, the dose-response relationships we have observed make it unlikely that the INSTI concentrations achieved in clinical practice would induce physiologically relevant antagonism of MC4R.

The regulation of body weight can also be modified via manipulation of other signaling pathways independent of MC4R. In nonclinical models, activation of the melanin-concentrating hormone receptor-1 [22], neuropeptide-Y receptors [23, 24] or ghrelin receptor [25] leads to weight gain. Whether INSTIs are able to activate these receptors is unknown. Furthermore, other MC4R-independent pathways are pharmacologically utilized to promote weight loss. One example is the glucagon-like peptide-1 (GLP-1) pathway, wherein synthetic peptide agonists of the GLP-1 receptor drive weight loss and are currently approved for the treatment of

diabetes and obesity [26]. In nonclinical models, blockade or loss of GLP-1 receptors does not promote weight gain [27], however it remains a possibility that INSTIs may affect the GLP-1 receptor or its ligands in a clinical setting to the extent it may increase body weight. These hypotheses remain to be directly tested. In summary, our findings demonstrate a class-wide ability of INSTIs to bind and antagonize MC4R only at very high drug concentrations, making the INSTI-MC4R interaction implausible as a molecular mechanism to explain INSTI-associated weight gain. Further work is needed to understand potential explanations for the observed association between INSTI use and weight gain.

## Supporting information

**S1 Fig. Example of antagonism response curve for INSTI EVG.** $EC_{50}$ value in the MC4R antagonism assay illustrated for EVG and positive control (SHU9119).
(TIF)

**S1 Table. Biochemical binding assay.**
(DOCX)

**S2 Table. Human $C_{max}$ values for approved INSTIs at recommended clinical dosages.**
(DOCX)

## Author Contributions

**Conceptualization:** Carrie McMahon, Kirsten White, Moupali Das, Hal Martin.

**Data curation:** Carrie McMahon, Kevin Holsapple, Kirsten White, Moupali Das.

**Formal analysis:** Carrie McMahon, James L. Trevaskis, Christoph Carter, Kirsten White, Sean Collins, Hal Martin, Leigh Ann Burns-Naas.

**Supervision:** Carrie McMahon, Leigh Ann Burns-Naas.

**Writing – original draft:** Carrie McMahon, James L. Trevaskis, Kevin Holsapple, Hal Martin.

**Writing – review & editing:** Carrie McMahon, James L. Trevaskis, Christoph Carter, Kirsten White, Moupali Das, Sean Collins, Hal Martin, Leigh Ann Burns-Naas.

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
