## [Decision Letter · Decision Letter 0]

29 Oct 2019

PONE-D-19-27424

Lack of an association between clinical INSTI-related body weight gain and direct interference with MC4 receptor (MC4R), a key central regulator of body weight

PLOS ONE

Dear Dr. McMahon,

Thank you for submitting your manuscript to PLOS ONE. After careful consideration, we feel that it has merit but does not fully meet PLOS ONE’s publication criteria as it currently stands. Therefore, we invite you to submit a revised version of the manuscript that addresses the points raised during the review process.

ADDITIONAL ACADEMIC EDITOR COMMENTS: There were 4 expert reviewers that handled your manuscript. All reviewers found this study interesting and it contains merit. However, there were several suggestions that would benefit the authors to include in the paper. There are suggestions to correct the figure labels/legends and include graphs of the data; there is information that needs to be added to the abstract and introduction; and in the discussion, future studies using in vivo models, which includes examining how other signaling molecules involved in food intake and body weight homeostasis might interact with this system, should included along with how these findings could be translated in the clinic.

We would appreciate receiving your revised manuscript by Dec 13 2019 11:59PM. To enhance the reproducibility of your results, we recommend that if applicable you deposit your laboratory protocols in protocols.io, where a protocol can be assigned its own identifier (DOI) such that it can be cited independently in the future. For instructions see: http://journals.plos.org/plosone/s/submission-guidelines#loc-laboratory-protocols

We look forward to receiving your revised manuscript.

Kind regards,

Frank T. Spradley

Academic Editor

PLOS ONE

Journal Requirements:

We note that one or more of the authors are employed by a commercial company: Gilead Sciences, Inc.

Reviewers' comments:

Reviewer's Responses to Questions

**Comments to the Author**

1. Is the manuscript technically sound, and do the data support the conclusions?

Reviewer #1: Yes

Reviewer #2: Yes

Reviewer #3: Yes

Reviewer #4: Yes

2. Has the statistical analysis been performed appropriately and rigorously? 

Reviewer #1: N/A

Reviewer #2: Yes

Reviewer #3: Yes

Reviewer #4: Yes

3. Have the authors made all data underlying the findings in their manuscript fully available?

Reviewer #1: Yes

Reviewer #2: Yes

Reviewer #3: Yes

Reviewer #4: Yes

4. Is the manuscript presented in an intelligible fashion and written in standard English?

Reviewer #1: Yes

Reviewer #2: Yes

Reviewer #3: Yes

Reviewer #4: Yes

5. Review Comments to the Author

Reviewer #1: In this study, several INSTIs were interrogated for their capacity for antagonism or agonism of MC4R in an in vitro cell-based assays including at concentrations far exceeding plasma

concentrations reached at the recommended dosages. The results indicate that while

INSTIs do exhibit the capacity to antagonize MC4R, this occurs at concentrations well

above predicted clinical exposure and is thus an implausible explanation for INSTIassociated

weight gain.

This is an interesting topic.

However, the discussion is very short and simplified and only two figures are shown in the main text.

Furthermore, I think more detail with regards to other studies investigating the effects on drugs/chemicals interacting the MC4R is interesting to include and discuss within the context of obesity.

E.g. the GLP-1 receptor analogue treatment and the MC4R both in cells, mice (eg. PubMed Central PMCID: PMC5550563.) and humans (eg PubMed PMID: 29861388) and many more...

From this it seems that weight loss can be induced even though you have no MC4R with GLP-1 RA. Thus obesity and MC4R seems complicated and treatment of obesity is not only dependent on the MC4R but also on other brain areas/receptors etc.

Furthermore actual and potential MC4R agonist treatment could be discussed.

Reviewer #2: McMahon and colleagues examined the impact of several integrase strand transfer inhibitors (INSTIs) on melanocortin-4 receptor (MC4R) activity. Given the recent increase in prevalence of excess weight gain in patients with HIV, particularly after initiating treatment with INSTIs, several hypotheses have been suggested to explain this weight gain including a potential interaction of INSTIs with the MC4R, leading to its inhibition promoting hyperphagia and reduced energy expenditure. Using in vitro assays in CHO cells transfected with human MC4R, the authors showed that although every INSTI compound exhibit significant binding to the MC4R and evoked inhibition of MC4R-induced activation of cAMP, the doses required for significant inhibition were far too greater than anticipated clinical circulating levels of free INSTI. Thus, the authors concluded that potential inhibition of MC4R by INSTIs is likely not the main mechanism behind the excess weight gain observed in INSTI-treated HIV patients.

Comments:

The manuscript is very well written, concise and easy to follow.

1) As already acknowledged by the authors, MC4R located in the brain, particularly in the hypothalamus, are responsible for body weight regulation and plasma levels of free INSTI may not necessarily reflect the concentration of INSTIs seen by MC4R in the hypothalamus. Also, all experiments were performed using in vitro techniques which may also not reflect 100% of the potential interaction of INSTIs with MC4R in vivo. How difficult would it be for the authors to include a separate protocol where MC4R KO animals (rats or mice) would be treated peripherally and/or intracerebroventricularly with INSTIs while their appetite, metabolic function and body weight are followed for a couple of weeks? I am not requiring the authors to perform such experiments, but they would significantly enhance the study's impact in the field.

Reviewer #3: In this paper, authors perform in vitro assay for some antiviral drugs effect on MC4R binding to explain its potental role in a side effect of the therapy associated with weight gain. The study found no agonistic effects of compounds and the antagonistic affects at concentrations which are much higher than blood level of these drugs in patients. The authors conclude that interactions of these drugs with MC4R is an unlikely mechanisms underlying weight gain in patients.

Specific comments

1. Abstract, specify that dolutegravir is an INSTI

2. In the introduction, please explain the chemical nature of the drugs

3. Is effect on appetite of INSTI known?

4. Result, can you provide graphs illustrating binding of both test and control compounds

5. Figure 1 legend is missing

6. Table 1, Agonists and antagonists columns have been switched between control substances

Reviewer #4: This paper investigates the effect of INSTIs on weight gain through MC4R. Despite INSTIs' capacity to antagonize MC4R, the authors have reported that at theurapeutic concentrations, antagonism of MC4R might not be an explanation for weight gain in treated patients. They are aware of the limitations which were mentioned.However, they need to speculate on what needs to be done from the research point more spesifically for practical purposes.

6. PLOS authors have the option to publish the peer review history of their article (what does this mean?). If published, this will include your full peer review and any attached files.

Reviewer #1: No

Reviewer #2: Yes: Alexandre Alves da Silva

Reviewer #3: Yes: Sergueï Fetissov

Reviewer #4: No

---

## [Author Response · Author response to Decision Letter 0]

24 Jan 2020

Reviewer #1: 

We thank the reviewer for this comment. Weight loss if, of course, extremely complicated and regulated by many central and peripheral pathways in homeostatic coordination. The first reference refers to evidence that liraglutide, a GLP-1 receptor agonist, can exert weight loss when infused directly to the brain and was associated with increased MC4R expression, thereby implicating that the weight loss may in part be due to melanocortin activity. The second paper refers to weight loss effects of liraglutide being equivalent in obese patients with or without MC4R inactivating mutations. The point is well taken, that additional mechanisms may be in play beyond MC4R. This is certainly true, and an additional paragraph has therefore been added to the discussion, where we discuss potential effects of agonism or antagonism of several pathways known to modulate food intake/body weight, at least in rodents. Very few pathways have been successfully targeted for weight loss therapy in a clinical setting. However, we list several of them as areas of potential future research on INSTI-associated weight gain.

Reviewer #2:

The authors appreciate the reviewer comments. The request for the potential evaluation of MC4R KO animals while treated with INSTIs was discussed among authors, however was considered outside of the scope of this body of work. The authors did add to the text the description of the lack of effect on body weight in repeat dose toxicology studies with bictegravir in rat, mouse or cynomolgous monkeys for dosing periods up to 39 weeks when administered daily. 

Reviewer #3:

The authors appreciate the suggested additions and/or revisions. The revised version with track changes reflect the list of recommendation. However, due to publicly available chemical structures for all INSTIs evaluated, the authors did not feel inclusion of the chemical structures was warranted in this case. Also, due to the sheer volume of graphs illustrating the biochemical binding of INSTIs evaluated with associated positive controls, the authors have chosen to include a representative figure for EVG only, which was associated with the lowest corresponding Cmax margin.

Reviewer #4: 

Thank you for your forward-looking comment recommending the inclusion of future research to be conducted. Please see response above to Reviewer #1.

---

## [Decision Letter · Decision Letter 1]

11 Feb 2020

Lack of an association between clinical INSTI-related body weight gain and direct interference with MC4 receptor (MC4R), a key central regulator of body weight

PONE-D-19-27424R1

Dear Dr. McMahon,

We are pleased to inform you that your manuscript has been judged scientifically suitable for publication and will be formally accepted for publication once it complies with all outstanding technical requirements.

With kind regards,

Frank T. Spradley

Academic Editor

PLOS ONE

Reviewer's Responses to Questions

**Comments to the Author**

1. If the authors have adequately addressed your comments raised in a previous round of review and you feel that this manuscript is now acceptable for publication, you may indicate that here to bypass the “Comments to the Author” section, enter your conflict of interest statement in the “Confidential to Editor” section, and submit your "Accept" recommendation.

Reviewer #2: All comments have been addressed

Reviewer #3: All comments have been addressed

Reviewer #4: All comments have been addressed

2. Is the manuscript technically sound, and do the data support the conclusions?

Reviewer #2: Yes

Reviewer #3: Yes

Reviewer #4: Yes

3. Has the statistical analysis been performed appropriately and rigorously? 

Reviewer #2: Yes

Reviewer #3: Yes

Reviewer #4: Yes

4. Have the authors made all data underlying the findings in their manuscript fully available?

Reviewer #2: Yes

Reviewer #3: Yes

Reviewer #4: Yes

5. Is the manuscript presented in an intelligible fashion and written in standard English?

Reviewer #2: Yes

Reviewer #3: Yes

Reviewer #4: Yes

6. Review Comments to the Author

Reviewer #2: No additional comments except that inclusion of in vivo studies in MC4R KO model would strengthen the manuscript.

Reviewer #3: (No Response)

Reviewer #4: (No Response)

7. PLOS authors have the option to publish the peer review history of their article (what does this mean?). If published, this will include your full peer review and any attached files.

Reviewer #2: Yes: Alexandre A. da Silva

Reviewer #3: Yes: Serguei Fetissov

Reviewer #4: Yes: Oya Ercan

---

## [Editor Report · Acceptance letter]

12 Feb 2020

PONE-D-19-27424R1 

Lack of an association between clinical INSTI-related body weight gain and direct interference with MC4 receptor (MC4R), a key central regulator of body weight 

Dear Dr. McMahon:

I am pleased to inform you that your manuscript has been deemed suitable for publication in PLOS ONE. Congratulations! Your manuscript is now with our production department. 

With kind regards,

on behalf of

Dr. Frank T. Spradley 

Academic Editor

PLOS ONE